# A Review of the Applications and Biodegradation of Polyhydroxyalkanoates and Poly(lactic acid) and Its Composites

**DOI:** 10.3390/polym13101544

**Published:** 2021-05-12

**Authors:** Jet Yin Boey, Lydia Mohamad, Yong Sen Khok, Guan Seng Tay, Siti Baidurah

**Affiliations:** School of Industrial Technology, Universiti Sains Malaysia, Minden 11800, Malaysia; jety31@gmail.com (J.Y.B.); lydiamohamad@student.usm.my (L.M.); pb5178@gmail.com (Y.S.K.); taygs@usm.my (G.S.T.)

**Keywords:** polyhydroxyalkanoates, poly(lactic acid), PHA-based composites, PLA-based composites, biodegradation

## Abstract

Overconsumption of plastic goods and improper handling of petroleum-derived plastic waste have brought a plethora of negative impacts to the environment, ecosystem and human health due to its recalcitrance to degradation. These drawbacks become the main driving force behind finding biopolymers with the degradable properties. With the advancement in biopolymer research, polyhydroxyalkanoate (PHA) and poly(lacyic acid) (PLA) and its composites have been alluded to as a potential alternative to replace the petrochemical counterpart. This review highlights the current synthesis process and application of PHAs and PLA and its composites for food packaging materials and coatings. These biopolymers can be further ameliorated to enhance their applicability and are discussed by including the current commercially available packaging products. Factors influencing biodegradation are outlined in the latter part of this review. The main aim of this review article is to organize the scattered available information on various aspects of PHAs and PLA, and its composites for packaging application purposes. It is evident from a literature survey of about 140 recently published papers from the past 15 years that PLA and PHA show excellent physical properties as potential food packaging materials.

## 1. Introduction

Plastic pollution has been one of the major concerns in environmental issues due to the massive production of single-use plastic driven by consumer demand, inevitably producing voluminous plastic waste. Petroleum-based plastics are non-biodegradable, and most of the plastic wastes are discarded to the environment or accumulated in landfills. Conventional petroleum-based plastics are recalcitrant to microbial degradation and accumulate in the environment and food chains [1]. Moreover, incineration of petroleum-based plastic waste for energy generation leads to the release of greenhouse-related and toxic gases, namely dioxin, furans and polychlorinated biphenyls. On the other hand, recycling can serve as a progressive alternative to manage plastic waste; however, recycling only works to limited extent, owing to the complex composition of plastic, types of plastic, material fatigue and quality loss with every recycling process. It is estimated that 9% of the plastic waste has been recycled [2,3]. Furthermore, 99% of the plastic was derived from non-renewable fossil hydrocarbon (petroleum and natural gas) [4].

In this modern lifestyle, the use of petroleum-based plastic is undeniably crucial in human daily life. It has been reported that two million tonnes of plastics were produced in 1950, and the annual production in 2015 had increased almost 200-fold, reaching 381 million tonnes [5]. Plastics are semi-synthetic organic polymers composed of long, chain-like molecules of high molecular weight [6]. Plastics are cheap, durable, resistant to degradation, and highly versatile, making them suitable to be used in a wide range of consumers and industrial applications. Despite the advantages possessed by petroleum-derived plastics, this has caused alarm with regard to large-scale pollutant [7].

Ethylene and propylene are the common monomers used to produce plastics, and none of the petroleum-derived plastics are biodegradable [5] because they are intractable to microbial degradation [6,8]. Consequently, rather than decompose, these plastics tend to accumulate in landfills, oceans, and the natural environment. Replacement of petroleum-derived plastics with bioplastic such as polyhydroxyalkanoates (PHAs) and poly(lactic acid) (PLA) has attracted much attention due to their beneficial properties, which are similar to conventional polymers and, additionally, they possess other important benefits such as biodegradability, biocompatibility, and non-toxic behaviour [9,10]. These unique properties of biopolymers gave rise to diverse applications in industries ranging from biodegradable packaging materials to biocompatible medical devices and tissue engineering [11,12].

In recent years, the increase in public awareness of plastic pollution and the rapid depletion of natural resources have become the main driving force for the development of biodegradable polymer from biological origin [13]. PHAs and PLA are among the most studied biopolymers as replacements for petroleum-derived plastics owing to their excellent physical properties and suitability as food packing especially their stretchability, toughness, and barrier properties. The main aim of this review article is to provide a summary of recent information concerning the synthesis process to obtain excellent physical properties of packing materials and biodegradation mechanism of PHAs and PLA, and its composites. The initiatives of utilizing biopolymers as packaging materials align with the United Nations Educational, Scientific and Cultural Organization (UNESCO) Sustainable Development Goal Number 12 to ensure sustainable consumption and production patterns.

## 2. Polyhydroxyalkanoates (PHAs) and Poly(lactic acid) (PLA): Overview and Synthesis Process

This chapter highlights the synthesis process for two types of biopolymer: PHAs and PLA. Synthesis of PHAs involves bacteria fermentation and extraction procedure. PLA is chemically synthesised via condensation polymerization of lactic acid (LA).

### 2.1. Synthesis of PHAs

#### 2.1.1. Fermentation

PHA is a bio-based polymer synthesized by various bacterial cells such as *Cupriavidus necator* and *Bacillus* sp. under specific growth conditions such as excess carbon sources and limited nitrogen sources [14,15]. PHAs were first discovered by Lemoigne in 1925 and are a family of naturally occurring biopolyester synthesized intracellularly by a great number of prokaryotes as storage polymers for carbon and energy sources [16,17]. Generally, PHAs can be synthesize via three pathways by utilization of various carbon sources: Pathway I is used predominantly in poly(hydroxybutyrate) (PHB) PHB-producing organisms, such as *C. necator* and *Bacillus* sp., while Pathways II and III are present in mcl-PHA producing *Pseudomonas* sp. [18,19,20]. Different monomers can be produced from various bacterial strains with different types of carbon substrate of the microorganisms and, therefore, co-polymer ratios can be tailored depending on its desired final product applications [17,21].

Research explores the utilization of various waste biomass as carbon sources for production of value-added products such as PHAs [10,22,23]. The aforementioned waste biomass requires pre-treatment to increase the corresponding monomer units for the ease of bacteria consumption. For example, molasses contain high amount of sucrose required to be hydrolyzed into its monomers glucose and fructose. An increment of 15% was observed for glucose and fructose content upon hydrothermal acid pretreatment of molasses [23].

*Cupriavidus necator*, formerly known as *Ralstonia eutropha*, is among the most studied PHB-producing strain due to its high PHB yielding capacity and it is mainly exploited at industrial scale [12,24]. By using a pure culture, this strain is able to accumulate intracellular reserves of PHB up to 80% of its cell dry weight (CDW) by consuming a broad range of carbon sources [24,25]. Typically, the molecular weight of PHB produced by wild-type bacteria is between 1.0 × 10^4^ to 3.0 × 10^6^ with a polydispersity index of about 2.0 [26]. To some extent, researchers have explored modifying the genetics of organisms to further enhance the PHAs with co-polymer production. High yield production of PHAs was reported by Kahar et al. with the utilization of recombinant strain PHB^−^4/pJRDEE32d13 (a PHA-negative mutant harboring *Aeromonas caviae* PHA synthase gene, *phaC_Ac_*) containing 5 mol% (*R*)-3-hydroxyhexanoate, P(3HB-*co*-5 mol% 3HHx), DCW of 128–138 g/L and a high PHA content of 71–74% (*w*/*w*). By utilizing wild-type strain *Ralstonia eutropha* H16, the CDW of 118–126 g/L and a high PHB content of 72–76% (*w*/*w*) were recorded [27]. The accumulated PHAs are then subjected to extraction procedures.

#### 2.1.2. Biological Extraction Method of PHAs

Extensively used chemical extraction methods of PHAs in industrial applications have caused a downside such as increased production cost, degradation of polymers, and environmental issues upon disposal. Hence, more attention and initiatives are being given to the biological extraction method to overcome these drawbacks. The idea of bio-extraction involves the use of living organism to extract the polymer from the cells.

There are several biological approaches that have been investigated. For example, by feeding the cells containing intracellular PHAs to the animals and insects [28,29,30,31]. Chee et al. demonstrated that up to 90% of PHAs can be recovered upon feeding the insects. The undigested biopolymer was excreted as fecal pellets. Further simple purification that does not involve hazardous chemicals are required to remove the impurities from the fecal pellets which will result in PHAs that have similar properties with the solvent-extracted ones [29,31].

Kunasundari et al. [28] has patented the use of Sprague Dawley rats to partially purify the PHB granules. The rats were given freeze-dried cells of *C. necator* H16 as a sole diet source for 7, 14, and 28 days. The mortality rate is zero throughout the study. The rats with bacterial cells diet produced low-odor white fecal pellets with PHA content of 87–90 wt%, while rats in the control group excreted normal blackish fecal pellets and no PHA was detected. The melting temperature (*T*_m_) and the degree of crystallinity (*X*_c_) of the biological recovered polymer obtained is within the average range and no significant deviation from the PHB was obtained by solvents [28].

Murugan et al. [29] has reported on the biological extraction of the polymer by utilizing the intestine of mealworms as an alternative approach to minimize the usage of solvents and chemicals and applicable on an industrial scale. It was found that mealworms readily ingested the freeze-dried cells of *C. necator* and whitish fecal pellets containing PHA were excreted out [29,30]. The purity of biologically extracted PHA washed with water is about 89% [29].

#### 2.1.3. Other Conventional Extraction Methods of PHAs

PHA is accumulated intracellularly in the form of granules in the bacterial cell cytoplasm [32]. Therefore, it is essential to break the cell wall in order to extract the PHAs. Effective PHA recovery from biomass components can be complicated and costly. It is estimated up to 50% of the polymer production costs is accounted from recovery process [26]. Numerous extraction and separation technologies have been developed on small scales as well as industrial scales. To facilitate cell disruption, a pre-treatment step is essential prior to the extraction by using chemical, physical or physicochemical means [26,33,34]. These processes include solvent extraction, chemical digestion, supercritical fluid disruption, enzymatic treatment, bead mill disruption, detergents, flotation techniques, use of gamma irradiation and aqueous two-phase system [33,34].

Table 1 shows the disadvantages and condition of conventional PHAs extraction methods together with the strains involved. Among the recovery methods available, solvent extraction by chloroform is the most preferred extraction method of PHAs. There are two steps involved in this solvent extraction method, first to break the cell wall using the solvent, and then to extract the polymer [26].

The PHAs from *C. necator* can be extracted by using chloroform as a solvent, with a ratio of cells to chloroform of 1:100 and stirring for five consecutive days [35]. Then, the separation of PHB is achieved by solvent evaporation or subsequent precipitation of the polymer from the non-PHB cell biomass in a non-solvent such as methanol or ethanol [26,33]. Chloroform extraction is not complex as compared to the other methods and very effective to separate the PHB granules [32]. Moreover, highly purified and high molecular weight of PHB can be recovered without the degradation of PHB molecules [30,32]. However, there are various disadvantages arising from the solvent extraction method. Solvent extraction involves the use of large volumes of toxic and volatile solvent [33,34] in order to modify the cell membrane permeability and to dissolve the polymer [26,30]. The PHB recovery from biomass using the solvent pre-treatment method has however raised an unfavourable concern to the environment and high consumption of solvent consequently increase the cost of the extraction process [26,33,34].

The chemical digestion method has also been commonly used in extracting the PHB from recombinant *Escherichia coli*. The biomass was mixed in sodium hypochlorite and treated at 30 °C for 1 h [36]. The purification method, however, possesses many drawbacks such as severe degradation of the PHB molecules, a large volume of wastewater produced, and treatment being needed in order to remove the surfactant from wastewater [33,34].

### 2.2. Advantages of Biological Extraction Method

Biological recovery of PHB granules accumulated in cells is preferably applied in a laboratory or on an industrial scale as compared to the conventional chemical extraction method. By utilizing the intestine of insects such as mealworm as a biological method to partially purify the PHB, the undesired drawback of the chemical extraction method can be avoided. The bio-extraction of PHB using insects is more relevant to employ at a large scale due to easy rearing and require minimal resources and space [15]. By utilizing the digestive system of insects as a green tool to extract the PHB from cells, this initiative can eventually reduce the production cost of PHB due to less solvents or chemicals being utilized.

This bio-extraction of PHB from cells is an alternative approach towards a green and sustainable method with the goal of minimizing the consumption of toxic solvents and strong chemicals [15,29]. The PHB polymer can be further purified in an eco-friendly method, by using water, sodium hydroxide or low concentration of surfactants such as sodium dodecyl sulphate and sodium dodecylbenzenesulfonate for pretreatment rather than chloroform and other harmful solvents [15,29,30].

It is interesting to note that the molecular weight (*M_w_*) of the biologically recovered PHB is comparable to the chloroform extracted PHB. This indicates that the biological extraction process did not degrade the molecular weight of PHB granules and was touted as the preferable extraction method [29,42]. In authors opinion, the usefulness of biological extraction method will be the subject of interest for many researchers due to the high PHA purity obtained and lesser amount of chemicals involved.

### 2.3. Synthesis of PLA

Poly(lactic acid) or polylactide (PLA), a biodegradable thermoplastic polyester, has gained attention as compared to other polyesters due to its potential to replace conventional petrochemical-based polymers [43] Good processability, sustainability and eco-friendly characteristics make PLA a favourable biopolymer, and it has thus gained attraction in fields such as packaging, textile, automotive composites, and biomedical application [44,45,46,47].

PLA is synthesized by direct condensation polymerization of lactic acid (LA) or by ring-opening polymerization (ROP) of lactide acid cyclic dimer, known as lactide [48,49]. Lactic acid is an organic acid which occurs naturally and can be produced by chemical synthesis or fermentation. Due to the prospects of environmental friendliness and using renewable resources instead of petrochemicals, interest in the fermentative production of lactic acid has increased. The carbon source for microbial production of lactic acid can be molasses, sugar cane bagasse, or whey, etc. [50,51]. In the polycondensation process, LA monomers are linked together through the reaction between the -OH and -COOH groups by removal of by-products, resulting to low molecular weight polymer [52].

Ring-opening polymerization (ROP) of lactide is used to produce a high molecular weight of PLA. First, the water is removed in a continuous condensation reaction of aqueous LA to produce low molecular weight prepolymers. Through internal transesterification, the prepolymer is then catalytically converted by ‘back-biting’ reaction to the lactide and purified. Three potential forms resulting from the production of cyclic lactide: D,D-lactide (called D-lactide), L,L-lactide (called L-lactide) and L,D- or D,L-lactide (called meso-lactide) [49]. The ratio and sequence of D and L-lactic acid units in the final polymer can be controlled depending on the monomer used and controlling reaction conditions [45].

However, PLA suffers certain drawbacks such as poor toughness, slow crystallization rate, and low heat distortion temperature [53,54]. The application of PLA could be extensive if its performance was enhanced to achieve suitable properties. Various approaches have been proposed, for instance, blending with other polymers, plasticizer [43,55,56,57], reinforcing materials in micro- (natural fibres, particles) and nanosize (nanoclays, carbon nanotubes, nanoparticles or nanocrystals) [50,53,58,59,60]. This review is mainly focused on blending PLA with polymers such as PHA, PLA/poly(butylene succinate) (PBS) and other polymers which are discussed further in the following sections.

## 3. PHAs-Based Biopolymer as Packaging Materials

### 3.1. PHB and P(HB-co-HHx)

PHB’s melting temperature (*T*_m_) is reported to be in a range of 173–180 °C, glass transition temperature (*T*_g_) is around 5 °C and high crystallinity degree of 40–60%. Various combination of co-polymer will produce biopolymer with various physical properties. For example, poly(3-hydroxybutyrate-*co*-3-hydroxyhexanoate [P(HB-*co*-HHx)] is extensively ameliorate for food packaging [61,62,63], consumables, medical devices and tissue engineering due to its mechanical properties, excellent biocompatibility and biodegradability [64,65,66]. PHB has similar physical properties as polypropylene, however, it is brittle and lack of toughness. Table 2 delineates PHA-based materials together with their advantages and disadvantages.

The initial mainstream applications of PHAs were the packaging and coating used in cosmetology and everyday articles, such as vials, bottles, and containers [67,68]. The PHAs based products was further expanded into disposable items and household goods, includes utensils, hygiene products and compostable bags [69]. In 2018, an Italian-based company, Bio-on SpA manufactured PHA microspheres designed to replace the conventional non-biodegradable micro-beads used extensively in cosmetics, body washes, as well as cleansers. The biodegradability of the PHAs microsphere can reduce microplastic pollution [67]. Several studies reported that strong PHB fibres with high tensile strength could be prepared by stretching the fibres after isothermal crystallization near the glass-transition temperature for 24 h and reactive extrusion with peroxide in a melt spinning process of PHB [82,83].

PHAs exhibit good barrier properties toward oxygen, carbon dioxide and moisture, which give them a potential to be used in food packaging [70,71]. PHB lamellar structure contribute to the superior gas barriers properties with vapor permeability approximately 560 g µm/m^2^/day, which is suitable for low-end food packaging applications [72,73].

Mulching is an important agriculture practice used to maintain good soil structure, moisture retention and weed control. The mulch films have been produced from Nodax™ P(3HB-*co*-3HHx) and Mirel™ PHB [74,75]. A patent by Havens et al. (2014) disclosed a modified fishing gear with a component made of PHA, which can biodegrade in an aquatic environment [76].

In terms of mechanical and thermal properties, the melting temperature (*T*_m_), degree of crystallinity (*X*_c_) and glass transition temperature (*T*_g_) of PHB are similar to PP, albeit stiffer and more brittle. The brittleness of PHB predominantly resulted from the formation of large crystals in the form of spherulite [84]. Owing to high *T*_m_ of PHB and thermal degradation occurred above its melting temperature, PHB has narrow processing window and make it low processable [85,86]. Doi and co-workers reported that P(HB-*co*-HHx) has a lower melting temperature, lower Young’s modulus, and higher elongation to break than PHB [87]. These indicate that P(HB-*co*-HHx) is tougher, more flexible and has a wider processing window than PHB. Although previous study reported that the physical properties of P(HB-*co*-HHx) was comparable to low-density polyethylene (LDPE), the physical properties of P(HB-*co*-HHx) are varied depend on concentration of comonomer unit and distribution of comonomer in the polymer chain [87,88,89,90]. The physical properties diversity of PHB and P(HB-*co*-HHx) widen the biotechnological application of PHAs in various industries, as well as influence the biodegradation of material during the end of product shelf life.

### 3.2. PHB/PLA Fibres

Due to the non-toxic and biodegradable nature, PHAs is considered a promising material to be utilized as coating in agriculture industry. The microencapsulated urea and a slow-release fertilizer that manufactured from the core-shell electrospun PHB/PLA fibres can control the timing and manner of the fertilizer delivery [77]. The biodegradable nature of PHAs make it a potential control release system for pesticides and herbicides. Several studies reported immobilized pesticides or herbicides can improve the pesticidal or herbicidal action and reduce the environmental toxicity [78,79].

### 3.3. Other PHA-Based Films

PHA-based films attract interest for food packaging applications due to non-toxicity, biodegradability and for exhibiting useful water vapour barrier properties. However, high production cost has limited its applications in food industry. There are several methods proposed to improve its economic viability which are blending with other biopolymers such as starch and PLA or fabrication of nanocomposite based on PHA and other organic or inorganic fillers, such as carbon nanotubes and cellulose nanowhiskers. The incorporation of other biopolymer or nano fillers can also further improve its crystallization behaviour, thermal stability and mechanical properties which are relevant from a food packaging perspective [61].

In addition, PHAs-based materials are used as a coating for paperboard in the food packaging industry [80]. In 2019, Danimer Scientific, a leading manufacturer of biodegradable plastic, partnered with UrthPact to manufacture drinking straw using Nodax™ PHA material. The company also announced a partnership with Genpak to manufacture a new line of food packaging, such as food containers [74,80].

## 4. PLA-Based Biopolymer as Packaging Materials

### 4.1. PLA-PHB

PLA has a significant demand in a wide spectrum of applications such as food coating, agricultural films, drug delivery systems and biomedical uses. A commercial PLA, trademarked Ingeo^®^ by NatureWorks LLC (Minnetonka, MN, USA), is used in market available products such as coffee capsules, diapers top-sheets or back-sheets, cups, yoghurt packaging, and electronics [1]. PLA is a growing alternative as a “green” food packaging polymer and has been used in the field of fresh products. The requirements for food packaging mainly evaluate their stretchability, toughness, and barrier properties (water vapour and oxygen permeability). Regulation of the atmosphere and moisture of food stored in the packaging allows external contaminations to be minimized and prolong its shelf-life [60,91]. Table 3 lists PLA-based materials together with their advantages and disadvantages.

With a view to increasing PLA compatibility and to maximizing their processing, poly(hydroxybutyrate) (PHB) and PLA blends were extensively studied as they have a similar melting temperature [91]. Arrieta et al. prepared PLA/PHB blends in 75:25 proportion, with the addition of plasticizer, poly(ethylene glycol) (PEG) and acetyl(tributyl citrate) (ATBC) [92]. The oxygen transmission rate values showed an improvement in barrier properties from PLA/PHB blend as compared to neat PLA films. PHB is able to increase the crystallinity of PLA and allowed more efficient molecular distribution in the polymer structure. Furthermore, PLA/PHB blend showed comparable tensile strength (TS) values to the PLA film due to the reinforcement effect, but less pronounced improvement in elongation at break. A similar finding was observed with the addition of 15 wt% PHB to the PLA matrix that contributed to a significant (*p* < 0.05) improvement in the oxygen barrier properties [93]. They reported no significant reduction in the water vapour permeability of neat PLA after blending with 15 wt% of PHB, despite the hydrophobic nature of PHB to these blends [93].

Ma et al. [94] developed biodegradable active packaging from PLA/PHB-based films containing 0.5% plasticizers, mono-caprylin glycerate (GMC) or glycerol monolaurate (GML), compared with EVOH-based film. The results highlighted that there was an improved oxygen permeabilities and better mechanical properties than those of EVOH-based film. In the preservative test, total bacterial counts (TBC) of salmon packed with PLA/PHB based film with GML as plasticizer reached 4.65 colony-forming units (CFU)/g on day 15; meanwhile, TBC of salmon dices sealed in other 2 films reached 6.65 and 6.35 CFU/g respectively on day 15, which exceeded the maximum permissible limit (6.0 CFU/g) for fish. Evinced PLA-PHB-based film with GML had higher WVTR than that of film formulation with GMC as plasticizer. This study indicated that PLA/PHB-based film with GML had better preservative effects and could be used as a viable alternative to non-biodegradable packaging for chilled salmon [94].

Jandas et al. [95] further increased the miscibility and flexibility of PLA-PHB by reactive compatibilization using maleic anhydride (MA). The PLA-PHB blend at ratio of 70:30 transformed from a brittle material to a ductile material for PLA-PHB-MA blends as a function of MA addition (from 1 wt% to 9 wt%), reaching the optimal flexibility of more than 500% by grafting 7 wt% of MA (PLA-PHB-MA). The increase in elongation suggests that there is some molecular interaction at the interface of PLA and PHB in the blend. Similarly, the biodegradation study revealed that PHB and nanoclay reinforcement enhanced the rate of PLA biodegradation. This has been confirmed by a weight loss study which reported that PLA/PHB/MA blend and PLA/PHB/MA/OMMT blend nanocomposites achieved 45% higher biodegradation rate than that of neat PLA. Factors for the enhanced biodegradability may be contribute to the presence of microbial based PHB within the matrix which enhanced the moisture absorption and molecular weight reduction that could contribute to improved biodegradability [95].

Jost et al. (2015) worked on PHBV/PLA blends prepared by solution casting to improve its miscibility and properties [96]. Merging of PHBV and PLA in 25:75 showed better miscibility behaviour while maintaining the biocompatibility of the material. The addition of 25 wt% PHBV resulted in an enhanced water vapour transmission rate to 46% and oxygen permeability to 40% in comparison to that of pure PLA, which are highly valued in the food-packaging industry.

### 4.2. PLA-PLA/Poly(butylene succinate) (PBS)

PLA/poly(butylene succinate) (PBS) blends and its nanocomposite with unmodified and modified cellulose nanocrystals (s-CNC) prepared by solvent casting were studied by Luzi et al. [97]. PLA and PBS pellet were solvent blended at three different weight ratios, adding 10, 20, 30 wt% of PBS into PLA solution. They reported that the combined addition of PBS and cellulose nanocrystals showed high barrier properties towards oxygen permeation, in which PLA with 20% PBS and 3% s-CNC represented the best formulation (oxygen permeability reduction of 47%). Additionally, overall migration tests with food stimulants were carried out to study the behaviour of potential food packaging in contact with the foodstuff. They found that the migration levels for all the studied bio-nanocomposites were maintained below the European legislative limits and touted as a suitable food packaging material [97].

Xie et al. established the introduction of PBS in PLA matrix to fabricate ductile PLA blends [98]. Blends of PLA/PBS were prepared by a melt blending technique, incorporating PBS at a content of 10 to 20 and 40 wt%. The unprecedented control of phase morphology and interfacial ligaments enhances interactions between the PBS nanofibrils and the PLA matrix. Consequently, the composites loaded with 40 wt% PBS nanofibrils demonstrated noteworthy material properties: higher strength, modulus and ductility as compared to those of pure PLA. Moreover, by adjusting PBS contents, the strength, stiffness, and ductility can be sufficiently tailored in a wide range, representing positive implications for expanding the application for PLA [98].

The plasticization effect exerted by acetyl tributyl citrate (ATBC) and isosorbide diester (ISE) on PLA-PBS based films was reported by Fortunati et al. [99]. By adding isosorbide diester (15 wt%), the PLA/PBS (80/20) blend shows a more homogeneous morphology, a drastically enhanced elongation at break (~250%) than unmodified PLA/PBS (80/20) blends (elongation at break: 10.5%). A lower cold crystallization and glass transition temperature were observed in the presence of the plasticizer, which increased the polymer flexibility, suggesting a potential application in the packaging sector [99].

Hongsriphan and Sanga [100] developed antibacterial food packaging by coating chitosan on biodegradable polymer sheets that were produced from melt blending PLA/PBS (90/10 wt%) using a simple dip-coating method. Surface modifications, corona treatment was applied to improve surface adhesion abilities. With the increment of the chitosan concentrations (0.25, 0.50, 1.0, and 2.0 *w*/*v*) coated on the PLA/PBS blend sheets, the reduction of both microorganisms (*S. aureus* and *E. coli*) could be observed. The antibacterial activity resulted from the chitosan coating on the substrates. Coating chitosan on sample sheets also reduced water vapour transmission rates compared to the non-coated ones. This can be explained by the strong intermolecular attraction (H-bonding) exhibited by chitosan, which provides a barrier layer against water vapour transmittance [100].

### 4.3. Other PLA-Based Composites

Songtipya et al. [101] explored the food packaging application of PLA/polybutylene adipate terephthalate (PBAT), which employed spent coffee ground and tea leaves as fillers. The study indicates that the overall migration values from all composite samples in all food stimulants did not exceed the maximum migration limit (10 mg/dm^2^), which comply with EU regulations. It is not possible for the chemicals to be diffused easily into food stimulants through the polymer composites. Therefore, PLA/PBAT composites used as food contact materials for packaging and containers have a minimum effect on food security [101].

Light transmittance plays an important role in agricultural film production, which influences the growth of plants as the photosynthesis of plants mainly absorb visible light at the wavelength of 400–700 nm. PDLA-g-PEG-g-PDLA(DPD) triblock polymer and Polyethylene glycol (PEG) are introduced to the PLLA matrix by melt compounding process [103]. The agriculture film have been produced from PLLA/DPD/PEG-10 film presented highly transparent (T% = 75.84%) with light transmittance property similar to PE (T% = 77.42%) under 700 nm light wavelengths. A remarkable improvement in mechanical properties was represented by PLLA/DPD/PEG-10 film which reached over 250% in elongation at break in both radial and axial directions compared with PLLA film. The film exhibits a rough surface that demonstrates a ductile fracture, indicating super toughness which makes it a good candidate for packaging and agricultural films.

## 5. Biodegradation Mechanisms

Extensive understanding and studies on the biodegradation mechanism of the biopolymers are required at the end of the product life cycle. As the potential applications of PHAs and PLA are widening, knowledge associated with the biodegradation mechanisms in different natural environments and controlled condition are important in order to predict the polymer behaviour in complex and dynamic environments [104]. Determination of the biodegradation can be achieved by several methods, which include physical methods (mass reduction of polymer, physical appearance of polymer, surface micro-morphology, reduction of molecular mass and strength properties), respirometric methods (gas evolution, biochemical oxygen demand) [105,106,107,108] and changes in chemical composition ratio by reactive-pyrolysis gas chromatography [109,110]. This chapter summarizes the biodegradation mechanisms for PHAs- and PLA-based composites.

### 5.1. Biodegradation of PHAs and PHA-Based Composites

The most attractive features of PHAs are their biodegradability and the non-toxic nature of degraded products. Biodegradation is defined as degradation of polymeric materials due to a cell-mediated phenomenon [111].

#### Biodegradation of PHB and P(HB-*co*-HHx)

Biodegradation studies of PHB and P(HB-*co*-HHx) have been conducted across a wide range of natural and controlled conditions, such as soil, mangrove swamp, river water, sea water, activated sludge, anaerobic sludge and compost aerobically or anaerobically [108,112,113]. Furthermore, studies of the biodegradation of PHAs are also conducted in vivo and in vitro, such as phosphate buffer, human blood, blood serum, muscular tissue of the rat and rabbit [114,115]. PHB is able to degrade rapidly in aerobic, anaerobic, and saline environments, hence it is easily disposed of without harm to the environment. In the case of biodegradation under aerobic condition, the end products of PHB and P(HB-*co*-HHx) are carbon dioxide and water, whereas under anaerobic condition, the degraded products are methane [112].

The biodegradability of PHAs is influenced by many factors, such as climatic factors and soil/water condition (moisture, temperature, pH, oxygen and nutrient availability), bacterial and fungal consortium, physiochemical properties of the polymer (surface area, chemical composition, molecular weight, degree of crystallinity, crystal lamellae) [112,116,117]. Incorporation of hexanoate monomer units into PHB polymer chains can enhance the physical properties and tailor the biodegradation rate. Previous studies reveal that the PHA biodegradation is dependent on the length of side chain. The PHAs with a longer side chain shows a high rate of biodegradability as compared to a short side chain PHAs [118]. Furthermore, Mergaert and co-workers reported that copolymers of PHAs has higher biodegradation rate than homopolymers [119].

P(HB-*co*-12%HHx) degrades faster than PHB in nutrient-depleted activated sludge due to lower crystallinity of P(HB-*co*-12%HHx) [107]. The authors reported that 40% of P(HB-*co*-12%HHx) was degraded after 18 days’ degradation in comparison with PHB, where only 20% was degraded. Electron microscopy analysis after biodegradation revealed that the biodegradation process (bioerosion) is initiated on the surface of the polymer by the PHAs depolymerase. The rough and porous surface of the polymer promote the bacterial colonization that secret PHAs depolymerase [107,120].

PHAs biodegradation is catalysed by intracellular and extracellular PHA depolymerases depending on the biophysical state of PHAs. Intracellular PHAs depolymerases are synthesized by the PHA-accumulating cells and hydrolyse or mobilize their own native PHA granules [121]. Native PHA granules exist in the mobile, amorphous state and surrounded by a layer of lipids and proteins. Conversely, extracellular PHA depolymerases are produced by other microorganisms to degrade the extracellular PHA polymer that is released by PHA-accumulating cells after death and cell lysis. Bacteria, fungi, algae and archaea are found to synthesize extracellular PHA depolymerases [112,122].

The depolymerization of PHAs involves two main steps. The first step involves adsorption of the binding domain of PHAs depolymerase onto the surface of the PHAs, while the second step involves hydrolysis of polyester chains by the catalytic domain of the enzyme [117]. The scissions of the polymer chains are initiated via end-scissions (randomly throughout the chain) and then by exo-scissions (from the chain ends). After hydrolysis, PHAs oligomers are produced and further depolymerised by oligomer hydrolase to produce organic acid. The microorganism utilizes the organic acid and further breakdown into carbon dioxide and water under aerobic condition [123]. The scission of the PHAs polymer chain eventually lead to a decrease in molecular weight, increase in dispersity and mass reduction [104].

### 5.2. Biodegradation of PHA Plastic Films in Soil

The biodegradation rate of PHAs was highly correlated to the PHAs molecular composition and degree of crystallinity of PHAs [110,117,124,125]. Conti and co-workers reported that PHB homopolymer is isotactic and has the most regular structure, resulting PHB has the highest crystallinity among the PHAs. The tight-chain packing region of the PHB crystalline region restricts the molecular mobility of the enzyme, resulting PHB has lower biodegradation rate than other PHAs [117]. Therefore, the incorporation of the 3HHx monomer into the PHB polymer chain disrupts the structure regularity of PHB as the 3HHx monomer is excluded from the crystalline phase of PHB and cannot co-crystallize in the same sequence of 3HB monomer which resulting in defect in the crystal lattice, increases amorphous region and reduce the crystallinity of polymer [125]. Furthermore, the microbial degradation of PHA occurs preferably at the amorphous end of the polymer rather than the crystalline region [126,127]. The increased amorphous region and low crystallinity of P(HB-*co*-HHx) enhance the biodegradation of the polymer. The 3HHx moieties in the polymer chain degrade in the preferential manner compared to 3HB moieties due to the 3HHx moieties decreasing with elapsed soil burial time [110]. Doi et al. reported that the crystallinity of P(HB-*co*-HHx) was significantly influenced by the 3HHx monomer content in the polymer [82].

Apart from crystallinity of the PHA films, the biodegradation rates of PHB and P(HB-*co*-HHx) are also influenced by the PHA depolymerases produced by the microbial population in the soil. Under complex microbial environments, various types of extracellular PHA depolymerase with different catalytic domains are secreted by various soil microorganisms to hydrolyse PHA. The enzymatic hydrolysis of different PHA films is greatly dependent on the substrate specificity of the active site in the catalytic domain of PHA depolymerase enzyme. The extracellular PHA depolymerases can be classified into 4 superfamilies with different catalytic structures and properties [128]. PHA will degrade faster in regions that have abundance of PHA-degrading microorganisms because there will be more contact between the microbes with the substrate to be degraded [105].

Several studies have reported the influence of soil types on biodegradation of PHAs. Kim et al. reported that 98.9% of PHB was degraded in activated sludge soil in 25 days whilst only 7.1% of PHB was degraded in forest soil [129]. Sridewi et al. (2006) reported that the PHA film buried in mangrove soil takes 4 weeks to reach 50% biodegradation [113]. Morse and co-worker reported P(HB-*co*-10%HHx) film reduced by 80% of its weight in 7 days when buried in a microcosm containing anaerobic digester biosolids [124]. In addition, the enzymatic hydrolysis of PHAs is known to be influenced by several factors, such as weather condition (temperature), soil condition (pH, moisture content, soil types, microbial population, burial depth), physiochemical properties of PHA (degree of crystallinity, crystal size and lamellar thickness, glass transition temperature, PHA monomer composition and distribution, end groups, geometry of polymer), and PHA processing technique (solvent cast film, pellet, electrospinning, extrusion) [105,116].

### 5.3. Biodegradation of PLA-Based Biopolymers

Biodegradability is one of the considerable aspects of sustainable materials. The fundamental mechanism involved in the biodegradation process of PLA blends is the action of microorganisms, such as algae, bacteria, and fungi on the polymer materials. In initial degradation step, PLA breaks down into lactic acid monomers, where the ester bonds are cleaved hydrolytically. Microorganisms further convert lactic acid into water and carbon dioxide [44]. Temperature, molecular weight, time, crystallinity and the amount of catalyst are the key factors affecting its degradation [130].

Polymer blends may have different morphologies based on blending ratios, which will influence the biodegradation process [131]. Biodegradation studies found that the addition of hydrophilic filler (starch and wood-flour) into PLA matrix accelerated the biodegradation rate higher than that of pure PLA in compositing testing [132]. The biodegradation rate increased from about 60% to 80% after 80 days with the starch content increased from 10% to 40%. The degradability enhancement may be attributed to the ability of natural fillers which facilitate water penetration into the PLA matrix more rapidly as well as bacteria growth. Both fillers speed up the thermal decomposition of PLA, with starch having a relatively stronger effect than wood-flour. When the filler content was increased to 40%, the decomposition temperature of PLA decreased by 40 °C. This may be due to small polar molecules being produced during decomposition of starch and wood-flour. The polyester chains in PLA are broken down in a similar fashion to break down by hydrolysis, causing a decrease in the decomposition temperature of PLA.

Arrieta et al. reported the mass loss as a measurement of the disintegration degree of plasticized PLA-PHB blends under composting conditions in a laboratory-scale test at 58 °C following the ISO-20200 standard [92]. The materials’ disintegrability started at polymers’ amorphous phase and was evidenced by the loss of transparency at the initial stage of the composting process. PHB acting as a nucleating agent decreased the PLA degradation rate by increasing the PLA crystallinity, since the ordered structure in the crystalline fractions could retain the action of microorganisms. By contrast, the plasticizers favoured the surface hydrolysis leading to substantial losses in mechanical properties, which also speeded up the disintegration phenomenon.

Persenaire et al. compatibilized poly(L-lactide) PLLA/PBS blends through a polyester maleation reaction using maleic-anhydride-grafted PLLA (PLLA-g-MA) and maleic-anhydride-grafted PBS (PBS-g-MA) [133]. The biodegradation rate was determined by accelerated weathering test, performed in a test chamber with conditions (50 °C, 50% RH) for up to 475 h. The results revealed that the addition of 20 wt% PBS to PLLA in the presence of compatibilizer allows the molar mass loss fraction to be doubled, demonstrating a promotion effect of PBS on the PLLA degradation. This observation can be related to the presence of homogeneously dispersed carboxylic acids produced by PBS degradation which favoured the hydrolysis of the PLLA matrix. PBS has a higher degradation rate than PLLA and, therefore, enhances its degradability when dispersed in the matrix.

The disintegration of PLA and PLA/PBS-based bio-nanocomposite reinforced with unmodified and surfactant modified cellulose nanocrystals have been studied in aerobic industrial composting conditions (58 °C, 50% RH) [97]. As a consequence of higher crystalline nature induced by PBS, blends of 20 wt% PBS content in PLA matrix showed a lower degradation rate with only 20–30% disintegratability while formulations reinforced with modified cellulose nanocrystals (s-CNC) demonstrated higher disintegration values (30–40%). Nevertheless, all samples have a degree of disintegration reaching 90% after 17 days.

## 6. Future Trends in Food-Packaging Materials

Food packaging plays a vital role in maintaining food quality and sensory properties, protecting the product against physical damage and minimizing food waste along the distribution chain (handling, transporting and storage). It is a key to preserve the food in order to maintain its quality and safety during its shelf life [134,135,136]. Non-biodegradable plastic materials including polypropylene (PP), polystyrene (PS), polyethylene (PE), and poly (vinyl chloride) (PVC) are known to be employed for packaging applications owing to their strong mechanical and physical characteristics [134,135]. Despite its importance andthe key role that packaging plays, it often ends up in landfill or the ocean, causing an environmental menace as it does not biodegrade upon disposal after use. As the public’s environmental awareness is gradually growing and there is a search for alternative material to replace petroleum-based plastics, the future of food packaging seems to lean towards sustainability to meet the preferences of its consumers.

In recent years, there has been growing interest in the development and industrial application of biodegradable polymers in terms of their inherent properties, environmental friendliness, and comparable mechanical and thermal properties. Among the biodegradable polymers, PLA is the most suitable nominees for future replacement of synthetic plastics, partly because it is approved by the U.S. Food and Drug Administration (FDA) as a food contact substance. PLA is transparent, non-toxic, economically feasible, commercially available and applications in the biomedical and agriculture field [137]. The production of PLA by fermentation of agricultural resources can reduce the emission of carbon dioxide and dependency on fossil resources [138]. However, the commercialization and various applications of PLA is limited due to their high production cost and brittleness. Currently, industrial and academic research has been undertaken to blend PLA with other biodegradable polymers to provide value added properties to the neat polymer without sacrificing its processability. Likewise, the physical properties and processability of PHAs can be further altered physically, chemically and enzymatically to enhance their performance in potential end-use application [139,140].

On average, PLA/PHB and PLA/PBS biocomposites have reached the level at which they can compete with the properties of fossil-based plastics and can be further exploited at the industrial level for packaging applications. Overall, important progress has already been achieved in terms of combining biodegradable polymer with PHAs and PLA portrayed by the large amount of available data from the literature. The use of bioplastic-based packaging allows significant advantages to be obtained in terms of environmental impact related to the entire life cycle of the product. For future perspectives, the post-consumption recycling of bioplastic-based systems can be carried out, as some commercial bioplastics do not undergo severe degradation under normal conditions. The degradation mechanisms must be reported as well for developing a basic understanding regarding the working procedure behind the degradation tests. Moreover, it is recommendable to investigate the potential migration of degradation products produced during either processing or biodegradation.

The food-packaging industry has gone through some considerable transformations in the 20th century. Many advancements in packaging technology appeared, such as intelligent or smart packaging (IOSP; time-temperature indicators (TTIs), gas indicators, microwave doneness indicators, radiofrequency identification (RFID), and others), and active packaging (such as oxygen scavengers, moisture absorbers, additives and antimicrobials) [101,135,141,142,143]. These innovations act by prolonging the shelf-life, preserving desirable food quality, providing indications about and regulating the freshness of food products. Further effort should concentrate on overcoming the technical constraints and high costs associated with these technologies, which have been the main factors restricting wider implementation and the development of additional commercial applications for new types of packaging material in the food-packaging industry. In future, we predict the innovative food packaging techniques to dominant the food packaging market due to their positive effects in solving ecological problems and increasing consumer acceptability.

## 7. Techno-Economic Challenges

Techno-economic challenges of the developed biopolymers often incur high production costs. The aforementioned hindrance should be addressed to enable the mass production of biopolymers. Various initiatives were explored by researchers such as modifying the genetics of microbes to enhance PHA production and utilizing ubiquitous biomass as carbon feedstocks. Moreover, low-cost biomass materials are currently being exploited either at laboratory or commercial scale for the sustainable production of PHAs and PLA due to its abundance and practicality. Furthermore, there is a bigger research scope for the feedstocks pre-treatment process. Amelioration seems to be the appropriate process in obtaining excellent and tailorable properties of biopolymer suitable for the final product application. In terms of future research perspectives, with the technology advancement and economic viability, we speculate an increase in patent filing and more industry sectors commercializing the biopolymers for packaging applications.

## 8. Conclusions

PHAs, PLA, and its composites have high potential to be applied as packaging materials. Over the years, the judicious utilization of various inexpensive and locally abundant biomass by-products has paved the ways towards a sustainable production of these biopolymers. Finally, we wish to comment on the biodegradation aspect. Most packaging materials should deem to possess biodegradable properties to some extent, either partially or, preferably, through complete biodegradation. Their many advantageous characteristics over recalcitrant petroleum-derived materials justify their use as eco-friendly materials.

## Figures and Tables

**Table 1 polymers-13-01544-t001:** Condition of conventional polyhydroxyalkanoate (PHA) extraction methods.

No.	Isolation Method	Strain	Condition of Extraction	Disadvantages	Purity (%)	References
1.	Solvent extraction	*C. necator*	Chloroform, ratio of cells to chloroform of 1 g:100 mL, stirring for five days	Consumption of large volume of toxic and volatile solvents Not environmentally friendly High capital and operation cost Difficulty in extracting PHA from solution containing more than 5% (*w*/*v*) PHB Native order of polymer chains in PHA granules might be disrupted Lengthy process Low recovery	95	[26,33,34,35]
2.	Chemical digestion	Recombinant *E. coli*	Sodium hypochlorite, 30 °C, 1 h	Low purity of PHA Large volume of wastewater Treatment needed to remove surfactant from wastewater Severe degradation of the polymer	93	[33,34,36]
3.	Supercritical fluid disruption	*C. necator*	Supercritical CO_2_, 100 min, 200 atm, 40 °C, and 0.2 mL of methanol	Dependent on process parameters Frequent need for clean up Difficulties in extracting polar analytes Difficulties in dealing with natural samples Low recovery	89	[33,34,37]
4.	Enzymatic treatment	*Sinorhizobium meliloti*	*Microbispora* sp. culture-chloroform	High cost of enzymes Complex process	94	[33,34,38]
5.	Bead mill disruption	*Alcaligenes latus*	Bead diameter of 512 µm, bead loading of 85%, 2800 rpm	Require several passes Long processing time Various process parameters have to be controlled precisely	-	[33,34,39]
6.	Flotation techniques	*Zobellella denitrificans MW1*	Chloroform, 30 °C, 72 h, self-flotation of cell debris overnight at room temperature	Consumption of large volume of toxic and volatile solvents	98	[33,34,40]
7.	Gamma irradiation	*Bacillus flexus*	Radiation doses of 10–40 kGy	Length of irradiation time High initial investment cost	-	[33,34,41]
8.	Aqueous two-phase system	*B. flexus*	Polyethylene glycol [PEG] 8000/phosphate, pH 8.0 and 28 °C, 30 min	Dependent on process parameters Issue of robustness and reproducibility Absence of commercial kits to evaluate aqueous two-phase system at bench scale Poor understanding of the mechanism	95	[33,34,41]

**Table 2 polymers-13-01544-t002:** Summarization of PHA-based materials.

No.	Application	Advantages	Disadvantages	References
1	Vials, bottles, and containers. (PHAs)	Biodegradable in both marine water and soil.	High production costs	[67,68]
2	Disposable items and household goods: utensils, hygiene products and compostable bags (PHAs)	Biodegradable.	High production costs	[69]
3	PHAs microspheres in cosmetics and body washes	Replacement for the conventional non-biodegradable micro-beads. Biodegradability of the PHAs microsphere reduces the microplastic pollution.	High production costs	[67]
4	Food packaging (PHAs)	Good barrier properties toward oxygen, carbon dioxide and moisture. Biodegradable.	High production costs	[70,71]
5	Food packaging (PHAs)	PHB lamellar structure contribute to the superior gas barriers properties with vapor permeability approximately 560 g µm/m^2^/day. Biodegradable.	High production costs	[72,73]
6	Mulch films [P(3HB-*co*-3HHx)]	Maintain good soil structure, moisture retention and weed control. Biodegradable.	High production costs	[74,75]
7	Mulch films (PHB)	Maintain good soil structure, moisture retention and weed control. Biodegradable.	High production costs	[74,75]
8	Modified fishing gear (PHAs)	Biodegradable in an aquatic environment.	Expensive	[76]
9	Microencapsulated urea and slow-release fertilizer (PHB/PLA fibres)	Non-toxic. Biodegradable. Control the timing and manner of the fertilizer delivery.	High production costs	[77]
10	Pesticide and herbicide	Improve the pesticidal and herbicidal action. Reduce the environmental toxicity.	High production costs	[78,79]
11	Food packaging (PHAs with organic or inorganic fillers)	Non-toxic. Biodegradable. Useful water vapour barrier properties. Improvement in terms of crystallization behaviour, thermal stability and mechanical properties.	Slightly lowered production costs.	[61]
12	Paperboard of food packaging (PHAs)	Non-toxic. Biodegradable.	High production costs	[80]
13	Drinking straw (PHAs)	Non-toxic. Biodegradable.	High production costs	[74,81]

**Table 3 polymers-13-01544-t003:** Summarization of PLA-based materials.

No.	Application	Advantages	Disadvantages	References
1	Food packaging (PLA/PHB blends 75:25 proportion)	Improvement in oxygen transmission rate	Less improvement in elongation at break	[92]
2	Food packaging (PLA/PHB blends 85:15 proportion)	Improvement in the oxygen barrier properties	-	[93]
3	Active packaging	Biodegradable Good tensile strength Improved oxygen permeabilities Better preservative effects	High water vapour permeability	[94]
4	Food packaging	Biodegradable High barrier properties towards oxygen permeation Overall migration levels are below the migration limits for food contact materials	-	[95]
5	Food packaging (PHBV/PLA blends in 25:75 proportion)	Improvement in the water vapour rate and oxygen permeabilities	-	[96]
6	Food packaging	Biodegradable High barrier properties towards oxygen permeation Overall migration levels are below the migration limits for food contact materials	-	[97]
7	Green packaging (PLA/PBS at PBS content of 10, 20 and 40 wt%)	Improved interfacial interactions Ductile Improvement in mechanical properties	-	[98]
8	Food packaging (PLA/PBS blends 80:20 proportion)	Homogeneous morphology Improvement in mechanical properties	Phase separation was observed with the plasticizer concentration of 20 wt%	[99]
9	Antibacterial food packaging (PLA/PBS blends 90:10 proportion)	Reduction of both microorganisms (*S. aureus* and *E. coli*) Good barrier layer against water vapour transmittance	Reduction in elongation at break values	[100]
11	Agricultural film (PLLA/DPD/PEG-10)	Highly transparent Good light transmittance Ductile	-	[101]
12	Food packaging and containers	Overall migration levels are below the migration limits for food contact materials	-	[102]

## Data Availability

The data presented in this study are available on request from the corresponding author.

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
