# Peer review of "A Review of the Applications and Biodegradation of Polyhydroxyalkanoates and Poly(lactic acid) and Its Composites"

_polymers, 2021, doi:10.3390/polym13101544_

Round 1

Reviewer 1 Report

The review article entitled ‘Polyhydroxyalkanoates, Poly(lactic acid) and Its Composites for Food Packaging Materials and Coatings: A review’. The concept of the review article is interesting and suitable to publish in Polymers Journal. However, in the present form it can not be accepted and it required substantial major revision.

The major comments are  as follows:

  • Title should be modified in a precise way.
  • Abstract looks very general and not informative should be rewritten. In abstract authors should mention the importance of research work briefly.
  • Provide a nice graphical abstract representing the overview of the MS with key highlights. For review article it should be mandatory and don’t use the figures used in the manuscript.
  • Introduction looks very general. In the introduction section, write the novelty of the work and the problem statement clearly. More discussion about the plastic pollution and their effects on the ecosystem Give quantitative data for this you can refer and cite the recent review article Bioresource Technology Volume 325, April 2021, 124685.
  •  Precise research objectives and clear justification of the selection this review topic is lacking thus major discussion is expected during revision.
  • PHAs can be synthesized by three pathways give details.  Regarding lignocellulosic biomass there are various studies including using water hyacinth waste biomass Polymers 12 (8), 1704, 2020, kenaf biomass Bioresource technology 282, 75-80, 2019 refer and cite their results during revision.
  • More importantly using waste biomass pretreatment is required add one section describing various methods used for pretreatment of various waste biomass refer recent important review article Fuel 251, 352-367, 2019.
  •  In Table 1 the author  discussed only disadvantages and what about advantages similarly this is also applicable for biological extraction methods what about disadvantages. In addition the author should give their opinion and view which is an important aspect.
  • Why  the major focus of the review article is on biodegradation? Instead the author should give more details about the different composites and their application.
  • Add one table describing the advantages and disadvantages in relation with commercial food packaging material.
  •  Techno Economic challenges of the developed composites need to be addressed Write the practical applications and future research perspectives by adding a new section before conclusions.
  • What are the limitations to use this methodology for commercial application?
  • The conclusion of the study needs to add with the specific output obtained from the study, it could be modified with precise outcomes with a take home message.
  • English and grammar mistakes are present. The author should check the manuscript by native English Speaker to improve the quality of the manuscript.

Author Response

Thank you for the input. Please refer the attachment for the revised version.

Reviewer 2 Report

After going through the manuscript " Polyhydroxyalkanoates, Poly(lactic acid) and Its Composites for Food Packaging Materials and Coatings: A review.", I would give my comments below.

- I think it's a parallel work with some new review papers that publish in recent months such as:
"Bugnicourt, E., Cinelli, P., Lazzeri, A., & Alvarez, V. A. (2014). Polyhydroxyalkanoate (PHA): Review of synthesis, characteristics, processing and potential applications in packaging.",

“Khosravi, A., Fereidoon, A., Khorasani, M. M., Naderi, G., Ganjali, M. R., Zarrintaj, P., ... & Gutiérrez, T. J. (2020). Soft and hard sections from cellulose-reinforced poly (lactic acid)-based food packaging films: A critical review. Food Packaging and Shelf Life, 23, 100429.”
and
"Gan, I., & Chow, W. S. (2018). Antimicrobial poly (lactic acid)/cellulose bionanocomposite for food packaging application: A review. Food packaging and shelf life, 17, 150-161."
There is not a new review manuscript for the present, so, what makes this review different from the others and from the most recent ones?

- Abstract should be rewritten. The general information should be concisely. Instead, more details of reviewed aspects should be presented.

- this manuscript needs more figures and comparative tables.

- Should be provided a comprehensive part between all of the Polyhydroxyalkanoates, Poly(lactic acid)  in the experimental and field-scale till now used. Add another table or tables.

- A review paper not only should summarize recently published works, but also should contain critical and comprehensive discussions. Therefore, check writing for the whole manuscript. The review should not be presented by listing what have done by others.
- Technical terms are misused through the manuscript and the writing needs a revision.

- Section of drawbacks and future could be increased quality of the manuscript.

- “PLA-based biopolymer as packaging materials” is written simply, most recent research and innovation in PLA performances should be reviewed to show the gap of knowledge. This part should be extended with recent research papers.

Author Response

(The authors gave the same response as above.)

Round 2

Reviewer 1 Report

The authors have substantially revised the manuscript according to the comments.

The present form of the manuscript can be accepted for publication.

Reviewer 2 Report

The manuscript quality has been improved.